# Transcriptome Analysis in Renal Transplant Biopsies Not Fulfilling Rejection Criteria

**DOI:** 10.3390/ijms21062245

**Published:** 2020-03-24

**Authors:** Francesc Moreso, Joana Sellarès, María José Soler, Daniel Serón

**Affiliations:** Nephrology Department, Hospital Universitari Vall d’Hebron, Department of Medicine, Autonomous University of Barcelona, Passeig Vall d’Hebron 119-129, 08035 Barcelona, Spain; jsellares@vhebron.net (J.S.); m.soler@vhebron.net (M.J.S.); dseron@vhebron.net (D.S.)

**Keywords:** renal transplantation, biopsies, transcriptome, microarrays, borderline changes, interstitial fibrosis and tubular atrophy

## Abstract

The clinical significance of renal transplant biopsies displaying borderline changes suspicious for T-cell mediated rejection (TCMR) or interstitial fibrosis and tubular atrophy (IFTA) with interstitial inflammation has not been well defined. Molecular profiling to evaluate renal transplant biopsies using microarrays has been shown to be an objective measurement that adds precision to conventional histology. We review the contribution of transcriptomic analysis in surveillance and indication biopsies with borderline changes and IFTA associated with variable degrees of inflammation. Transcriptome analysis applied to biopsies with borderline changes allows to distinguish patients with rejection from those in whom mild inflammation mainly represents a response to injury. Biopsies with IFTA and inflammation occurring in unscarred tissue display a molecular pattern similar to TCMR while biopsies with IFTA and inflammation in scarred tissue, apart from T-cell activation, also express B cell, immunoglobulin and mast cell-related genes. Additionally, patients at risk for IFTA progression can be identified by genes mainly reflecting fibroblast dysregulation and immune activation. At present, it is not well established whether the expression of rejection gene transcripts in patients with fibrosis and inflammation is the consequence of an alloimmune response, tissue damage or a combination of both.

## 1. Introduction

Renal transplantation is the best treatment for end-stage renal disease. During the last decades the advent of new immunosuppressive therapies has been associated with a decline in the incidence of T-cell mediated rejection (TCMR), while antibody-mediated rejection (ABMR) has emerged as one of the main reasons for graft deterioration. Graft function monitoring relies on serum creatinine, estimated glomerular filtration rate (eGFR) and urinary protein excretion. Unfortunately, these parameters have low accuracy to predict graft outcome and the renal biopsy is still the gold standard to evaluate graft dysfunction and establish prognosis. In addition to indication biopsies, some centers perform surveillance biopsies to detect subclinical inflammation and quantify the progression of chronic lesions. However, low inter- and intra-observer reproducibility constitutes a limitation of histological evaluation [1] and an objective molecular approach offers an opportunity to improve diagnostics and better characterize the nature of these conditions.

The borderline changes category defined by the Banff consensus has an uncertain significance. While it can represent true TCMR, it also can be the morphological correlate of acute kidney injury associated with ischemia-reperfusion or other types of injury. Criteria for its diagnosis have been changing over time; in the first Banff meeting, this category included biopsies with mild tubulitis (t1) associated with mild to severe interstitial inflammation (i1–i3) [2]. In the 2005–2007 meetings, the classification was updated, and renal transplant biopsies showing mild to severe tubulitis (t1–t3) without interstitial inflammation (i0) were also included into this category [3]. Recently, evidence suggests that patients with biopsy scoring ≥ i1t1 have a poorer outcome than patients with i0t1 bringing into question the current diagnostic thresholds [4]. Furthermore, therapeutic approaches toward borderline changes vary in different clinical settings. The diagnosis of borderline changes in biopsies caused due to graft dysfunction leads to anti-rejection treatment with steroid pulses, while no treatment is given in most centers when this lesion is observed in surveillance biopsies. In a large clinic-pathological study conducted in more than 500 patients monitored with indication and surveillance biopsies, borderline changes emerged as a heterogeneous diagnostic group, ranging from mild inconsequential inflammation (resolved in about 60% of untreated cases) to clinically significant TCMR able to induce immune-mediated tubular injury [5].

Interstitial fibrosis and tubular atrophy (IFTA) in indication kidney transplant biopsies is a common finding and it is associated with different active diseases: ABMR, TCMR, glomerulonephritis, polyoma virus infection or pyelonephritis. However, a proportion of indication and surveillance biopsies only show IFTA with variable degrees of inflammation with no other associated lesions. Surveillance biopsies displaying IFTA with inflammation in non-fibrotic areas (IFTA+i) have a shortened graft survival in comparison with biopsies only displaying IFTA or inflammation [6,7,8,9,10,11]. Additionally, graft survival is shortened in indication or surveillance biopsies with inflammation in areas of interstitial fibrosis (i-IFTA) [12,13]. Of note, this histological phenotype is associated with minimization of immunosuppression [14]. Altogether, these findings have inspired the definition of chronic active TCMR as the presence of inflammation in areas of fibrosis (i-IFTA ≥2 and t ≥ 2), which has been included in the latest Banff classification [15]. However, molecular characterization of borderline changes, as well as of other histological phenotypes characterized by the presence of inflammation and fibrosis in healthy or fibrotic interstitium, is incomplete. Molecular profiling to evaluate renal transplant biopsies using microarrays has been shown to be an objective measurement that adds precision to conventional histology [16]. Here, we review the contribution of transcriptomic analysis in surveillance and indication biopsies with borderline changes and IFTA associated with variable degrees of inflammation.

## 2. Transcriptomes in Biopsies Categorized as Borderline Changes

Seminal studies conducted with reverse transcription polymerase chain reaction (RT-PCR) have shown that the quantification of the gene expression of the cytotoxic T-cell molecules (perforin, granzyme B and Fas ligand) either in blood, urine or renal biopsies, is associated with acute rejection [1,7,8,9,10,11,12,13,14,15,16,17,18,19]. Moreover, it has been proposed that the granzyme B transcript expression level could discriminate between non-rejecting grafts, borderline changes and tubule-interstitial rejection grade IA [20]. Similarly, molecular data demonstrate that histological rejection found in protocol biopsies from stable, well-functioning grafts, can be associated with an increase in immune-activation gene expression, similarly to acute clinical rejection [21,22]. Later, different experimental and clinical studies showed that the histologic lesions diagnostic of TCMR are associated with, but not mediated by, perforin or granzyme, indicating that the granule-associated cytotoxic mechanisms of T cells are not the effectors of TCMR [23]. At the beginning of this century, the advent of parallel gene analysis with microarrays provided a method for large-scale human gene transcription analysis [24]. Sarwal et al. [25] were the first to apply the genome-wide technology using microarrays in biopsies of rejecting kidney allografts, confirming the expression of T-cell transcripts while other studies were not able to distinguish molecular rejection from other types of injury [26,27]. Gene expression constitutes a complex biological process occurring across different diseases in kidney allografts. In order to facilitate the interpretation of this process, different relevant genes have been studied, leading to the generation of gene lists denominated pathogenesis-based transcripts sets (PBTs) derived from mouse [28,29,30,31], human germinal cell lines [32] and the literature [33,34]. The PBTs correlate strongly with one another and with histopathology in biopsies for clinical indications, and these molecular changes reflect biological processes that are common across different types of injury, including rejection. These patterns include the expression of T-cell, macrophage and IFN-gamma-induced transcripts, as well as PBTs reflecting the injury-repair response [34,35,36].

Reeve et al. distinguished rejection from non-rejection using predictive analyses of microarrays in an unselected biopsy for a cause population in which the full spectrum of diseases was represented [37]. This study showed that TCMR and ABMR shared many transcripts, mainly those involving IFN-gamma effects (e.g., CXCL9, CXCL11, GPB1), since both types of rejection can trigger release of IFN-gamma: TCMR through T cells and ABMR through NK cells [36]. Although these IFN-gamma inducible transcripts are present in both alloimmune diseases, they are not only specific for rejection and can be found in other forms of injury [38,39,40]. Biopsies with minimal PBT disturbances had a very low incidence of rejection, but disparities between histological and molecular diagnosis was found in some cases. In patients treated for rejection before biopsy, histopathologic diagnosis of rejection was associated with a low molecular rejection score. In addition, biopsies with borderline changes and severe IFTA were associated with a high rejection score [35]. Of note, the PBTs were not different in lesions scored t1 versus t2 (the threshold between borderline changes and TCMR), confirming the low reproducibility of the t1–t2 distinction observed by pathologists. Relatively low PBT scores were observed in some biopsies with intimal arteritis with minimal tubulo-interstitial changes that did not meet the Banff criteria for TCMR based on the i and t lesions (i < 2, t < 2). In this regard, in a study aimed to explore the utility of the molecular diagnosis in indication biopsies displaying borderline changes, it was shown that most cases designated borderline by histopathology (67%) were found to be non-rejection by molecular phenotyping [41]. This finding is consistent with a clinical study in which borderline biopsies behaved as a very heterogenous group, ranging from mild non-progressive inflammation to overt TCMR [5]. These results suggest that some current Banff histopathology criteria may be unreliable, particularly at the cut-off between borderline changes and T-cell mediated rejection.

The INTERCOM study aimed to evaluate the potential impact of microarray diagnosis of TCMR in kidney transplants [42]. This prospective multicenter study generated a microarray-based TCMR score in a population of 300 indication biopsies of kidney transplants, using an algorithm developed in a reference set. The TCMR score was able to reclassify 77/300 biopsies (26%), including 46 borderline biopsies that were reclassified as TCMR score positive (*n* = 8) or negative (*n* = 38) [42]. These results demonstrated that the TCMR score offers new insight in biopsies categorized as borderline for their reclassification.

Our group also participated in a study that confirmed the feasibility of real-time central molecular assessment and offered a new and standardized dimension in allograft biopsy analysis [43]. This study showed that the borderline category is usually no rejection, and it can be associated with acute kidney injury and inflammatory primary renal diseases, suggesting that the treatment in these cases should not be focused on renal allograft rejection [43]. In contrast, Hrubá et al. observed an increase in immunity and inflammation genes in biopsies for clinical indication with borderline changes performed early in comparison with borderline 3-month protocol biopsies [44]. This increase observed within two weeks of transplantation may be influenced by the initial alloimmune response and by the ischemic injury as well. Moreover, specific transcripts were also associated with graft function deterioration. Therefore, similar histological findings reflect wide variations in the gene expression profiles associated with borderline diagnosis [44]. The differences between these two studies may be in part be ascribed to the huge variation in the timing of a biopsy (1423 vs. 7 days) [43,44]. This may suggest that the significance and prognosis of borderline histology in renal allografts seems to be worse when the allograft biopsy is performed during the first week after transplantation. To further evaluate these discrepancies, new studies with a larger sample size and with new methods to analyze data have been conducted. Microarray data from 1208 kidney transplant biopsies collected prospectively from 13 centers were analyzed by a cross-validation classifier score predicting the presence of ABMR, TCMR and five related histologic lesions generated using supervised machine learning methods. Biopsies classified as borderline by histology (*n* = 109) were classified with this approach as no rejection (72%), TCMR (6%), ABMR (20%) or mixed ABMR/TCMR (1%). It is important to remark that a huge discrepancy also exits for biopsies classified by histology as TCMR (*n* = 87) since only 43% were classified as TCMR, while 31% were classified as no rejection and 13% as ABMR [45]. A recent study also addressed this discrepancy by comparing molecular and histology diagnoses in kidney transplant biopsies and reported similar findings [46]. In Table 1, the main findings of the described studies in indication biopsies are summarized.

Transcriptome analysis in surveillance biopsies performed during the first year after transplantation has allowed to characterize molecular and cellular mechanism triggered by the ischemia-reperfusion injury and by the alloimmune response as well as the injury-repair mechanisms leading to progressive fibrosis [47,48,49]. The relationship between subclinical tubulo-interstitial inflammation (borderline changes and TCMR) and progressive scarring in protocol biopsies has offered contradictory results. In a cohort of protocol biopsies performed at 6 weeks (*n* = 107), it was observed that biopsies diagnosed as borderline changes (*n* = 20) or TCMR (*n* = 9) have an increased expression of T-cell, macrophage and IFN-gamma-induced transcripts and a decreased expression of kidney transport transcripts, but PBTs correlated with a delayed graft function and not with progression of IFTA at six months or renal function decline at two years. These results suggest that the molecular phenotype reflects the injury-repair response to implantation stresses and has little relationship with future events [50]. The Genomics of Chronic Allograft Rejection (GoCAR) study analyzed the transcriptome in 3-month surveillance biopsies and described that in patients with subclinical rejection (20 out of 154 patients) there was a progressive scarring at 12 months, either evaluated by the Chronic Allograft Damage Index or by the Banff scores (ci+ct). However, exclusion of patients with subclinical or borderline rejection did not alter the ability of the gene set to predict higher chronic damage, suggesting that inflammation was not the main driver for the derivation of the gene set [51]. Recently, a signature in peripheral blood to characterize patients with subclinical borderline changes and subclinical TCMR in 3-month protocol biopsies was described in a GoCAR study and validated in an independent cohort. In this study, patients with subclinical rejection (borderline changes *n* = 35, TCMR *n* = 11) had a significantly higher risk of clinical acute rejection at 12 and 24 months, faster decline in graft function, faster progression of chronic lesions and decreased graft survival than patients without rejection (*n* = 145). By RNA sequencing on whole blood they identified a 17-gene signature that accurately diagnosed subclinical/borderline TCMR [52]. From a clinical point of view, it has been shown that the combination of tacrolimus, mycophenolates and steroids is associated with the lowest prevalence of subclinical rejection (TCMR type 1 and borderline rejection) [8,53,54,55], but its contribution to progressive renal scarring has not been documented in some studies [50,55]. In Table 2 the main findings in surveillance biopsies are summarized.

Thus, the transcriptome changes occurring in well-functioning grafts after transplantation monitored by surveillance biopsies have been properly described, but the contribution of the ischemia-reperfusion injury and the specific alloimmune response to these changes has not been completely elucidated. In this regard, immune response in rejection is to a certain extent independent of the type of transplanted organ [56] and these transcriptomic changes are also observed in patients with infection, cancer or autoimmune diseases [57]. In summary, available studies suggest that the probability to have a molecular diagnosis of rejection is higher in indication than in surveillance biopsies displaying borderline changes. However, the implication of this finding regarding treatment deserve further studies.

## 3. Transcriptome Analysis in Biopsies Categorized as Interstitial Fibrosis/Tubular Atrophy

Fibrosis is a hallmark of many diseases that can be triggered by different stimuli. It can appear spontaneously or as a response to tissue damage, inflammatory diseases, cancer or to foreign implants. Irrespective of its primary cause, fibrosis is nearly always associated with inflammation [58]. In kidney transplants, IFTA with inflammation is associated with reduced graft survival [8]. However, the main driving force of fibrosis in kidney transplants is not always clear. It has been associated with ischemia reperfusion injury and with most common diseases leading to graft failure, such as humoral rejection, cellular rejection, glomerulonephritis or polyoma virus infection; however, fibrosis with inflammation not clearly associated with an identifiable cause constitutes a common finding.

Microarray studies performed in stable grafts have shown that gene expression is a dynamic process that varies in a time-dependent manner. A study evaluating differentially expressed genes between 1, 3 and 12 months surveillance biopsies and implantation biopsies, showed that immune activity peaked after one month, fibrotic-associated gene expression at 3 months while the wound healing and repair process remains activated between 3 and 12 months. At 12 months, macrophage-related genes were overexpressed [47]. Accordingly, as previously mentioned the interpretation of gene expression in stable grafts must take into consideration the timing of the biopsy.

In a study evaluating 46 patients with a 12-month surveillance biopsy not showing TCMR or ABMR, a positive correlation was observed between the number of infiltrating macrophages and IFTA. Furthermore, biopsies with IFTA > 0 showed 739 differentially expressed genes in comparison to patients without fibrosis. Most differentially expressed genes were macrophage-associated genes (CD48 and CCL2) and T cell IFN-gamma-associated genes (CXCL9 and CXCL10). The top pathways associated with fibrosis were T-cell antigen presentation and T-cell toxicity [59]. In another 12-month surveillance biopsy study, including 86 normal biopsies, 45 biopsies with fibrosis and 20 biopsies with fibrosis and inflammation, it was described that IFTA with inflammation was associated with reduced graft survival and increased transcripts associated with Toll-like receptor (TLR) signaling, antigen presentation, dendritic cell maturation, IFN-gamma, T cell cytotoxic and acute rejection-associated gene pathways, suggesting that some gene pathways activated in patients with IFTA and inflammation are shared with rejection and reflect the presence of infiltrating cells. On the other hand, in this study, some subtle differences between fibrosis without inflammation and normal histology were observed as increased IL6, vascular cell adhesion molecule 1, L-selectin, alternative macrophage activation and IFN-gamma-associated genes. However, the five-year allograft survival in patients with IFTA without inflammation was similar to patients with a normal histology [9]. In a study including indication and surveillance biopsies, Modena et al. [60] characterized gene expression in patients with IFTA associated with TCMR, with inflammation and without inflammation. Gene expression in IFTA with TCMR and with inflammation was similar to the rejection biopsies. An unexpected result was the observation that in IFTA without inflammation, 80% of differentially expressed genes were shared with rejection. In a study comparing patients with subclinical rejection with or without IFTA, it was observed that activation of IL10 and increased infiltration by B lymphocytes was characteristic of patients with inflammation and fibrosis despite a similar degree of cellular activation in both groups [61].

Gene transcripts in biopsies with inflammation in areas of IFTA are different from biopsies with inflammation in healthy areas. In an indication biopsy study, it was observed that T cell, IFN-gamma, macrophage and a set of injury-repair transcripts were overexpressed in biopsies with inflammation in healthy areas. On the other hand, B cell, immunoglobulin, mast cell and a different set of injury-repair transcripts were overexpressed in biopsies, mainly displaying inflammation in scarred tissue. There was nearly no overlap between gene expression in biopsies with inflammation in healthy areas and inflammation in areas of fibrosis [62]. In sequential surveillance biopsies IFTA progressed during the first year [63]. In indication biopsies done during the first year there is a time sequence of gene expression: injury transcripts peak during the first days after transplantation, fibrillar collagen during the first weeks and immunoglobulin and mast cell transcripts at four months. In this study, some inflammatory pathways also correlate with fibrosis, such as IL1-beta, TNF, IFN-gamma and TGF-beta1. However, the stronger correlation between injury kidney transcripts and fibrosis than with inflammatory pathways led the authors to propose that fibrosis may reflect recent and active kidney wounding as ischemia reperfusion injury or other active diseases [36,64]. In another study of unselected indication biopsies, it was observed that the molecular diagnosis of TCMR in i-IFTA biopsies was uncommon, while the closest correlation with i-IFTA was acute kidney injury transcripts, challenging the notion proposed at the 2017 Banff meeting that moderate to severe i-IFTA constitutes an histological phenotype of chronic TCMR [65].

IFTA has also been associated with enhanced expression of growth factors, including the hepatocyte growth factor (HGF), epidermal growth factor (EGF), vascular endothelial growth factor (VEGF), mitogen activated protein kinase signaling, integrin pathway [66], metzinins, zinc-dependent metalloproteinases that regulate extracellular matrix composition [67] or different miRNA’s, such as miR-142, which participates in the regulation of macrophages and T-regs or miR-2014 that regulates apoptosis [68].

Different studies have attempted to evaluate genes associated with progression of IFTA. Scherer et al. [48] evaluated differential gene expression in 3-month normal surveillance biopsies that progressed to IFTA at six months in comparison to patients without progression. They described that before progression to IFTA, there was increased immune activation and fibroblast-associated genes. Other groups have confirmed these results [47,49]. Modena et al. evaluated 40 biopsies with IFTA without inflammation and classified them according to the immune and inflammatory gene expression. Patients with high expression had a reduced renal allograft survival [60]. Finally, O’Connell et al. described that the 13 genes that are mainly expressed in fibroblasts allow to predict the progression of fibrosis at 1 year, as well as graft survival [51]. These data suggest that IFTA displays variable degrees of immune activation and dysregulation of fibrotic genes that allow to stratify patients at risk of progression of fibrosis and graft loss.

After transplantation, there is a variable and sequential activation of rejection-associated cell metabolism, cell growth and repair transcripts. This activation depends on the severity of the ischemia-reperfusion injury, alloimmune response and the repair ability of the renal tissue. However, a detailed picture of this sequence of transcriptome changes has not been precisely defined. In Figure 1 we hypothesize the sequence of events leading to fibrosis with or without inflammation.

## 4. Conclusions

Transcriptome analysis applied to biopsies with borderline changes allows to distinguish patients with rejection from those in whom mild inflammation mainly represents response to injury. Biopsies with IFTA and inflammation mainly occurring in healthy tissue display a molecular pattern that is similar to T-cell mediated rejection while biopsies with IFTA and inflammation mainly occurring in scarred tissue, apart from T-cell activation, also express B cell, immunoglobulin and mast cell-related genes. Additionally, it has been shown that patients at risk for IFTA progression can be identified by a gene set mainly reflecting fibroblast dysregulation and immune activation. At present, it is not well established whether the expression of rejection gene transcripts in patients with fibrosis and inflammation is the consequence of alloimmune response, tissue damage or a combination of both.

## Figures and Tables

**Figure 1 ijms-21-02245-f001:**
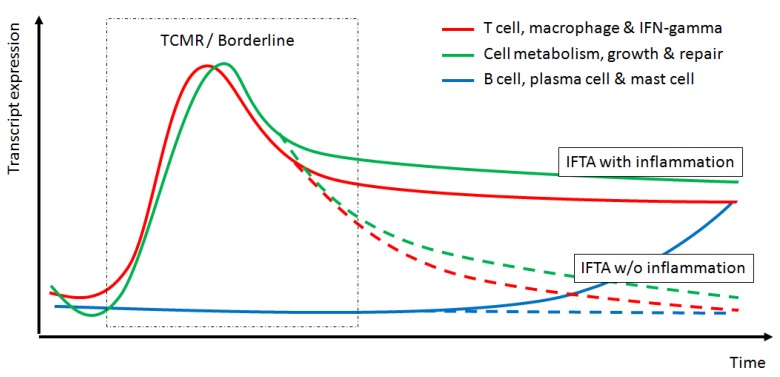
Hypothesized sequential modifications in the transcriptome for different gene sets and its correlation with histological changes. TCMR, T-cell mediated rejection; IFTA, interstitial fibrosis and tubular atrophy. Continuous lines depict transcriptome expression leading to IFTA with inflammation while dotted lines depict transcriptome expression leading to IFTA without inflammation.

**Table 1 ijms-21-02245-t001:** Transcriptome analysis on indication biopsies with borderline changes-histological category and different controls. ABMR, antibody-mediated rejection; TCMR, T-cell mediated rejection; BL, borderline changes; PBTs, pathogenesis-based transcripts; CATs, infiltration of cytotoxic T cells; GRIT1, interferon-gamma and rejection induced transcripts; KT, kidney parenchymal transcripts; IFTA, interstitial fibrosis and tubular atrophy; AKI, acute kidney injury; BL, borderline changes.

Reference	Time & Type of Biopsy Sample Size	Methods & Results	Main Conclusion
Mueller TF et al.	Clinical indication	Affymetrix GeneChip Human Genome U133 Plus 2.0 Arrays	ABMR and TCMR manifested similar PBT disturbances. Biopsies with minimal PBT disturbances had a very low incidence of rejection.
Am J Transplant 2007 [35]	*N* = 143.	↑ CAT1, CAT2, GRIT1, GRIT2;	
		↓ KT1-KT2 in TCMR GRIT1 associated with C4d staining (ABMR)	
De Freitas DG et al.	Clinical indication	Affymetrix GeneChip Human Genome U133 Plus 2.0 Arrays	Most cases designated borderline by histopathology are found to be non-rejection by molecular phenotyping.
Am J Transplant 2011 [42]	TCMR(*n* = 35), BL (*n* = 45), non-rejection (*n* = 116)	Molecular changes measured according to T-cell burden; a rejection classifier; a canonical TCMR classifier; and the risk score. Reassigned borderline biopsies as TCMR like 13/40 (33%) or non-rejection-like 27/40 (67%).	
		Decision tree analysis showed that i-total >27% and tubulitis extent > 3% match the molecular diagnosis of TCMR in 85% of cases.	
Halloran PF et al.	Clinical indication	Affymetrix microarrays.	The molecular TCMR score has potential to add new insight, particularly in situations where histology is ambiguous or potentially misleading.
Am J Transplant 2013 [43]	International Collaborative Microarray Study (*n* = 300).	Microarray expression files for BFC403 (GSE36059) and INT300 (GSE48581) cohorts. TCMR scores divided into high or low using the same cut off of 0.1.	
	TCMR (*n* = 32), BL (*n* = 46)		
Hrubá P et al.	BL early clinical biopsies (*n* = 13) and 3-month protocol biopsies (*n* = 15)	Illumina microarray analysis.	Variations in gene expression between clinical and subclinical borderline changes despite similar histological findings.
Kidney Int 2017 [45]		↑ C19orf59, CXCL2, IL6, S100A8, S100A9, FGA in early clinical biopsies as compared to protocol biopsies	
		↑ SAA1, CLEC5A, FGA in borderline biopsies with IFTA progression	
Halloran PF et al.	Clinical indication	Affymetrix hgu219 PrimeView microarray chips.	MMDx would add valuable support for clinical decisions beyond current standard-of care.
Am J Transplant 2017 [44]	International Collaborative Microarray Study (*n* = 519).	Molecular classifier scores (ABMRpm [positive ≥0.20], TCMRt [positive ≥0.10], Rejection [positive ≥0.30])	
	ABMR (*n* = 88), ABMR suspected (*n* = 10), TCMR (*n* = 29), AKI (*n* = 43), BL (*n* = 31), atrophy/fibrosis (*n* = 84), “no abnormalities” (*n* = 141).		
Reeve J et al.	Clinical indication. 13 centres (*n* = 1208)	Affymetrix hgu219 PrimeView microarray chips.	Borderline changes are classified as no rejection (72%), TCMR (6%), ABMR (20%) and mixed ABMR/TCMR (1%).
JCI insights 2017 [46]	BL (*n* = 109), TCMR (*n* = 87)	Archetypal analysis of molecular phenotypes.	

**Table 2 ijms-21-02245-t002:** Summary of different studies analyzing the transcriptome on surveillance biopsies. IFTA, interstitial fibrosis and tubular atrophy; TCMR, T-cell mediated rejection; QCATs (infiltration of cytotoxic T cells); GRIT1 (interferon-gamma and rejection-induced transcripts); QCMAT (infiltration of macrophages; AMAT1 (alternative macrophage activation); IRITD3 (injury and repair induced transcripts); ENDATs (endothelial transcripts); KT1 and KT2 (kidney parenchymal transcripts).

Reference	Time & Type of Biopsy Sample Size	Methods & Results	Main Conclusion
Sherer A et al. Nephrol Dial Transplant 2009 [49]	Paired 3- and 6-month protocol biopsies.	Affymetrix GeneChip Human Genome U133 Plus 2.0 Arrays	Gene expression profiling of early protocol biopsies identified changes in the transcriptome of grafts, which may be important for development of IFTA.
	Non IFTA progression (*n* = 12)	IFTA progression is associated with overexpression of T-, B-cell activation, immune response and profibrotic genes.	
	IFTA progression (*n* = 8)	Under expression of genes related with transporter and metabolic functions in IFTA progression.	
Vitalone MJ et al.	Paired 0-, 1-, 3- and 12-month protocol biopsies (59 biopsies from 18 patients)	Human 8K cDNA microarrays, Australian Genome Research	Allografts display immune and fibrotic gene expression profiles with patterns of expression gradually varying by time after transplantation.
Transplantation 2010 [48]	Subclinical rejection = 14%	Immune pathway activity peaked at 1-month, fibrotic expression at 3 months, wound healing-remodelling and cell proliferation-repair processes were activated between 3 and 12 months, whereas macrophage-related gene expression occurred late by 12 months	Gene expression predated histologic damage.
Naesens M et al.	Paediatric transplants.	Affymetrix Gene Chip Human Genome U133 Plus 2.0 Arrays	Progressive chronic histological damage is associated with regulation of both innate and adaptive immune responses that cannot be evaluated by histology.
Kidney Int 2011 [50]	24 patients with paired 0-, 6- and 24 months protocol biopsies. 24 patients with TCMR.	Upregulation of adaptive (T- and B-cell signatures) and innate immune cell transcripts (dendritic cell and NK cell transcripts) is already present in biopsies of kidneys several months before chronic histological damage occurs.	
Mengel M et al.	6-weeks protocol biopsies (*n* = 107). TCMR (*n* = 9), Borderline (*n* = 20)	Affymetrix Gene Chip Human Genome U133 Plus 2.0 Arrays	The molecular phenotype reflects the injury–repair response to implantation stresses, and has little relationship to future events.
Am J Transplant 2011 [51]		↑ QCAT, QCMAT, GRIT1, AMAT; ↓ KT1-KT2 in TCMR and borderline.	
		PBTs correlated with DGF but not with ΔeGFR at 2 years, ΔIF/TA at 6 months or i-Banff at 6 months.	
O’Conell PJ et al.	Discovery set: 3-month (*n* = 159) and 12-month paired protocol biopsies (*n* = 101).	Affymetrix human exon 1.0 ST array in the discovery set and qPCR in the validation set. 13 genes related with active repair and regeneration pathways predicts the development and progression of chronic allograft damage and subsequent allograft loss	Kidney transplant recipients at risk of allograft loss can be identified before the development of irreversible damage.
The Lancet 2016 [52]	Validation set (*n* = 45)		

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
