# Peer review of "Transcriptome Analysis in Renal Transplant Biopsies Not Fulfilling Rejection Criteria"

_ijms, 2020, doi:10.3390/ijms21062245_

Round 1

Reviewer 1 Report

Dear authors:  You have tackled and important area in the field of kidney transplantation  It appears that the main goal of your paper is to review the literature with the aim of discerning acute rejection from the wide ranging findings of borderline rejection.  There are a couple issues that need to be addressed:

  1. There are some grammar and syntax errors that make reading the manuscript challenging. 
  2. Organization of the manuscript. The first paragraph of the introduction is good. The concepts in the introduction lay a good foundation but the grammar makes it difficult to piece together.
  3. Do you have a reference for 60% of the inconsequential inflammation resolved?  Was this with or without intervention
  4. It was difficult to differentiate between the findings in the section on transcriptomes in borderline rejection and T cell mediated rejection. Is there a way to make this a more clear picture of the implications of these different profiles? There seems to be some difference in the expression with IFTA based on your reports.
  5. It might be worthwhile contemplating whether it makes a difference whether a biopsy is done for cause or surveillance. The findings are should be clinically handled regardless of why the biopsy was performed.
  6. It would be helpful to your readers to develop a table of what the implication is of these different trancriptome profiles are related to the histologic finding. Which profiles are associated with progressive fibrosis or acute tissue injury. At the end of reading the manuscript twice it was difficult for me to state what the take home message would be and whether microarrays would be worth pursuing. I believe the data is there in the manuscript but it is difficult to sort out.
  7. There is some valuable message to be conveyed with this manuscript but the organization needs to be restructured to make the story more clear.

Author Response

You have tackled and important area in the field of kidney transplantation  It appears that the main goal of your paper is to review the literature with the aim of discerning acute rejection from the wide ranging findings of borderline rejection.  There are a couple issues that need to be addressed:

  1. There are some grammar and syntax errors that make reading the manuscript challenging. 

Grammar and syntaxes have been reviewed.

  1. Organization of the manuscript. The first paragraph of the introduction is good. The concepts in the introduction lay a good foundation but the grammar makes it difficult to piece together.

We have reviewed the grammar of the introduction and we have done some modifications of the text in order to facilitate its understanding.

  1. Do you have a reference for 60% of the inconsequential inflammation resolved?  Was this with or without intervention

Reference has been clarified. This refers to untreated borderline inflammation as it is described in the reference 5.

  1. It was difficult to differentiate between the findings in the section on transcriptomes in borderline rejection and T cell mediated rejection. Is there a way to make this a clearer picture of the implications of these different profiles? There seems to be some difference in the expression with IFTA based on your reports.

We appreciate this comment since we realized that these two sections have some overlap that difficult the understanding of the main message. In the new version we have merged both parts in one single section.

  1. It might be worthwhile contemplating whether it makes a difference whether a biopsy is done for cause or surveillance. The findings are should be clinically handled regardless of why the biopsy was performed.

In the section entitled “Transcriptome in biopsies categorized as borderline changes” we separately review molecular diagnosis in indication and surveillance biopsies. At the end of this section we have added the following sentence: “Available studies suggest that the probability to have a molecular diagnosis of rejection is higher in indication than in surveillance biopsies displaying borderline changes. However, the implications of this finding with regard to treatment deserve further studies”

  1. It would be helpful to your readers to develop a table of what the implication is of these different transcriptome profiles are related to the histologic finding. Which profiles are associated with progressive fibrosis or acute tissue injury. At the end of reading the manuscript twice it was difficult for me to state what the take home message would be and whether microarrays would be worth pursuing. I believe the data is there in the manuscript but it is difficult to sort out.

Figure 1 has been added in order to summarize the relationship between rejection, injury-repair transcripts and IFTA with or without inflammation.

  1. There is some valuable message to be conveyed with this manuscript but the organization needs to be restructured to make the story clearer.

The organization of the paper has been modified in order to facilitate its reading.

Reviewer 2 Report

In this manuscript, the authors give the wide and well-documented review of the utility of transcriptome analysis applied to both surveillance and protocol kidney graft biopsies. Such an additional analysis aimed to help with more precise distinction between borderline changes suspicious for T-cell mediated rejection (TCMR) or interstitial fibrosis and tubular atrophy (IFTA) with interstitial inflammation. For instance, the transcriptome analysis applied to biopsies with borderline changes allows to distinguish patients with rejection from those in whom mild inflammation mainly represents response to injury. Additionally, patients at risk for IFTA progression can be identified by genes mainly reflecting fibroblast dysregulation and immune activation. They conclude that, to date, it is not well established whether the expression of rejection gene transcripts in patients with fibrosis and inflammation is the consequence of an alloimmune response, tissue damage or a combination of both.

 General comment:

There are many typing errors, especially in the second half of the manuscript, including extra spaces, and the notorious lack of “γ” sign standing by IFN, i.e. interferon gamma.  

Minor comments:

Line 77-78: “stereotyped” is not a right word here, I think. Usually? Standard?

Line 139: should be „meet” instead of „me”

Table 1: the description of GRIT1 is incomplete, whereas BL description is given twice

Line 188: should be probably “tubulo-interstitial”

Author Response

Reviewer 2

In this manuscript, the authors give the wide and well-documented review of the utility of transcriptome analysis applied to both surveillance and protocol kidney graft biopsies. Such an additional analysis aimed to help with more precise distinction between borderline changes suspicious for T-cell mediated rejection (TCMR) or interstitial fibrosis and tubular atrophy (IFTA) with interstitial inflammation. For instance, the transcriptome analysis applied to biopsies with borderline changes allows to distinguish patients with rejection from those in whom mild inflammation mainly represents response to injury. Additionally, patients at risk for IFTA progression can be identified by genes mainly reflecting fibroblast dysregulation and immune activation. They conclude that, to date, it is not well established whether the expression of rejection gene transcripts in patients with fibrosis and inflammation is the consequence of an alloimmune response, tissue damage or a combination of both.

 General comment:

There are many typing errors, especially in the second half of the manuscript, including extra spaces, and the notorious lack of “γ” sign standing by IFN, i.e. interferon gamma.  

a, b and g Greek symbols have been substituted by the expressions alpha, beta and gamma expressions and typos have been corrected.

Minor comments:

Line 77-78: “stereotyped” is not a right word here, I think. Usually? Standard?

Stereotyped has been omitted in the new version.

Line 139: should be „meet” instead of „me”

This error has been corrected

Table 1: the description of GRIT1 is incomplete, whereas BL description is given twice

Description of GRIT1 has been completed and borderline changes has been abbreviated as BL in table 1.

Line 188: should be probably “tubulo-interstitial”

This typo has been also corrected.